# Novel imaging biomarkers for mapping the impact of mild mitochondrial uncoupling in the outer retina *in vivo*

Bruce A. Berkowitz[1]*, Hailey K. Olds[1], Collin Richards[1], Joydip Joy[1],
Tilman Rosales[1], Robert H. Podolsky[2], Karen Lins Childers[2], W. Brad Hubbard[3,4],
Patrick G. Sullivan[3,4,5], Shasha Gao[6,7], Yichao Li[6], Haohua Qian[6], Robin Roberts[1]

1 Department of Ophthalmology, Visual and Anatomical Sciences, Wayne State University School of
Medicine, Detroit, MI, United States of America, 2 Beaumont Research Institute, Beaumont Health, Royal
Oak, MI, United States of America, 3 Spinal Cord and Brain Injury Research Center, University of Kentucky,
Lexington, KY, United States of America, 4 Department of Neuroscience, University of Kentucky, Lexington,
KY, United States of America, 5 Lexington VA Health Care System, Lexington, KY, United States of America,
6 Visual Function Core, National Eye Institute, National Institutes of Health, Bethesda, MD, United States of
America, 7 Department of Ophthalmology, First Affiliated Hospital, Zhengzhou University, Zhengzhou, China

☯ These authors contributed equally to this work.
* baberko@med.wayne.edu

UNITED STATES

**Data Availability Statement:** All relevant data are
within the manuscript and its Supporting
Information files.

## Abstract

### Purpose

To test the hypothesis that imaging biomarkers are useful for evaluating *in vivo* rod photore-
ceptor cell responses to a mitochondrial protonophore.

### Methods

Intraperitoneal injections of either the mitochondrial uncoupler 2,4 dinitrophenol (DNP) or
saline were given to mice with either higher [129S6/eVTac (S6)] or lower [C57BL/6J (B6)]
mitochondrial reserve capacities and were studied in dark or light. We measured: (i) the
external limiting membrane–retinal pigment epithelium region thickness (ELM-RPE; OCT),
which decreases substantially with upregulation of a pH-sensitive water removal co-trans-
porter on the apical portion of the RPE, and (ii) the outer retina R1 (= 1/(spin lattice relaxation
time (T1), an MRI parameter proportional to oxygen / free radical content.

### Results

In darkness, baseline rod energy production and consumption are relatively high compared
to that in light, and additional metabolic stimulation with DNP provoked thinning of the ELM-
RPE region compared to saline injection in S6 mice; ELM-RPE thickness was unresponsive
to DNP in B6 mice. Also, dark-adapted S6 mice given DNP showed a decrease in outer ret-
ina R1 values compared to saline injection in the inferior retina. In dark-adapted B6 mice,
transretinal R1 values were unresponsive to DNP in superior and inferior regions. In light,
with its relatively lower basal rod energy production and consumption, DNP caused ELM-
RPE thinning in both S6 and B6 mice.

**Funding:** This research was gratefully supported by the National Institutes of Health [RO1 EY026584 (BAB) and R01 AG058171 (BAB)], Kentucky Spinal Cord and Head Injury Research Trust (KSCHIRT) Grant 14-13A and VA Merit Award 1I01BX003405 (PGS), NIH intramural Research Programs EY000503 and EY000530 to HQ, NEI Core Grant P30 EY04068, and an unrestricted grant from Research to Prevent Blindness (Kresge Eye Institute, BAB), a Fight for Sight Summer Student Fellowship (CR), and Wayne State University School of Medicine Medical Student Summer Research Fellowships (HKO and JJ).

**Competing interests:** The authors have declared that no competing interests exist.

## Conclusions

The present results raise the possibility of non-invasively evaluating the mouse rod mitochondrial energy ecosystem using new DNP-assisted OCT and MRI *in vivo* assays.

## Introduction

Measuring regulation of the mitochondrial energy ecosystem is thought to be useful for indicating vulnerability to metabolic challenges as well as an index of treatment efficacy. Mitochondria function is often studied by measuring responses in either isolated mitochondria, cells, or excised tissue following metabolic stimulation via a mitochondrial uncoupler, such as 2,4-dinitrophenol (DNP), [1–3]. Exogenous uncouplers, or endogenous uncoupling proteins, shuttle protons across the mitochondrial inner membrane thus disrupting the mitochondrial proton gradient used to generate ATP. *Ex vivo* studies suggest that mitochondrial protonophores stimulate metabolism but only if sufficient reserve capacity is available [3–5]. For example, if mitochondrial reserve capacity is relatively limited, then DNP is not expected to increase respiration by much with correspondingly little change in the production of oxygen free radicals, and $CO_2$ and waste water [3, 6, 7]. On the other hand, neurons with a relatively larger mitochondrial respiratory reserve capacity exposed to DNP will show relatively larger reductions in their local levels of oxygen with greater flux through the electron transport chain. In turn, this results in a larger reduction in the leakage of damaging oxygen free radicals from the mitochondria; $CO_2$ and water production are also expected to increase with higher metabolic rate leading to greater water removal in, for example, the outer retina [5, 6, 8]. Importantly, as recently reviewed, preclinical studies indicate that low doses of DNP can be useful as safe neuroprotective agents in a range of diseases [9]. However, there is a need for new imaging biomarkers with translational potential for evaluating the impact of mitochondrial protonophores *in vivo* [10].

Rod photoreceptor mitochondria utilize their spare respiratory reserve to respond to energy challenges, oxidative stress, and degeneration [1, 2]. Thus, in this study, we examined two novel imaging biomarkers for non-invasively mapping outer retina responses following metabolic stimulation via mild mitochondrial uncoupling with DNP. First, OCT was used to assess external limiting membrane–retinal pigment epithelium (ELM-RPE) thickness as a surrogate of localized hydration status [4, 11, 12]. For example, we have reported that physiologic mitochondrial respiratory changes with dark-and light-adaptation have a measurable impact on ELM-RPE thickness and water mobility as measured by OCT and diffusion MRI *in vivo*; notably, OCT showed a higher detection sensitivity likely due to its order-of-magnitude greater spatial resolution [4, 11, 12]. In this study, we predict that light-dependent metabolic stimulation with DNP will increase $CO_2$ and waste water production with an increase in water removal in / thinning of the ELM-RPE region, as shown by dark-evoked constriction of the ELM-RPE region and suggested previously by DNP-evoked reduction of edema in a nerve injury model [5, 8, 11–13]. Second, MRI measurements of layer-specific R1 [= 1/(spin lattice relaxation time (T1)] were performed since R1 is expected to decrease with, for example, decreased levels of paramagnetic oxygen and / or production of paramagnetic free radicals expected with DNP [14–20].

Here, we test both predictions by comparing DNP responses on OCT and MRI in 129S6/eVTac (S6) and C57BL/6 (B6) mice. Baseline rod photoreceptor oxygen consumption rate is greater in B6 mice than in S6 mice, as measured *ex vivo* [4]. These studies take advantage of

the fact that low doses of DNP can be used safely in animals, and that rod cells are the dominant photoreceptor in the outer retina with mitochondrial-dense inner segments in mouse retina [9, 10, 21, 22].

## Materials and methods

All animals were treated in accordance with the National Institutes of Health Guide for the Care and Use of Laboratory Animals, the Association for Research in Vision and Ophthalmology Statement for the Use of Animals in Ophthalmic and Vision Research, approval by the Wayne State University Institutional Animal and Care Use Committee, and approval by the National Eye Institute Animal Care and Use Committee. We used male 2–3 month old 129S6/EvTac (Taconic labs) or C57BL/6 (Jackson Labs) mice. Animals were housed and maintained in 12 hour:12 hour light-dark cycle laboratory lighting until the day before the experiment. After the experiment, mice were humanely euthanized by cervical dislocation followed by a bilateral pneumothorax, as detailed in our IACUC-approved protocol.

### DNP dosing

DNP from Sigma–Aldrich, St. Louis, MO, USA was used [9, 23]. R1 data were collected as described below from dark-adapted mice given either saline (vehicle, equal volume), or 5 or 10 mg DNP/kg intraperitoneal (IP) at ~1 hr before MRI examination; ELM-RPE thickness were measured under similar conditions but after administration of either saline or 10 mg DNP/kg IP at ~1 hr before OCT testing.

### Optical Coherence Tomography (OCT)

For comparison to MRI, OCT (Envisu R2200 VHR SDOIS) was used on the left eye to visualize retinal layer spacing *in vivo* in subgroups of mice (n = 2 per group) as previously described. Mice were anesthetized with urethane (36% solution intraperitoneally; 0.083 ml/20 g animal weight, prepared fresh daily; Sigma-Aldrich, St. Louis, MO). 1% atropine sulfate was used to dilate the iris, and GenTeal was used to lubricate the eyes. Since representative OCT images are being compared to averaged MRI data, location of two anatomic landmarks are approximate, and key boundaries are indicated by dashed vertical lines. We have generated a body of work that supports the alignment of the vitreous-retina (0% depth) and retina-choroid (100% depth) borders of OCT to MRI images (see for example [24]).

For measuring ELM-RPE thickness, ultrahigh resolution OCT was used as previously reported [4, 13, 25, 26]. Briefly, dark-adapted mice were kept in dark over-night (~16 hrs), and light-adapted mice were exposed room light for ~5 hrs. Mice were anesthetized with (100 mg/kg) and xylazine (6 mg/kg), retina OCT images were captured by the same operator (SG) with Envisu UHR2200 (Bioptigen, Durham, NC, USA) first as baseline and the next day under the same condition 1 hr after injection of DNP. Mice received saline injection and imaged with the same protocol showed no change on OCT image (data not shown). The left mouse eye was positioned with the ONH in the center of the OCT scan. Full field volume scans (1.4 mm × 1.4 mm at 1000 A-scan × 100 B-scan × 5) and two radial scans (at horizontal and vertical position and averaged 40 times) were captured. Averaged radial scan images were used for retinal thickness measurement. Measurements were performed on 4 spots (450 μm from center of ONH at both horizontal and vertical directions) by using the vendor-provided Reader program (Bioptigen), and an averaged number was used as the measurement for the eye. Outer retina length was measured from the external limiting membrane to the RPE-choroid boundary as previously described [13].

## MRI

Mice were kept in darkness for at least 16 hrs before, as well as during, the MRI examination because this condition maximizes photoreceptor mitochondrial activity [27–29]. In all groups, immediately before the MRI experiment, animals were anesthetized with urethane (36% solution IP; 0.083 mL / 20 g animal weight, prepared fresh daily; Sigma–Aldrich, St. Louis, MO, USA) and treated topically with 1% atropine followed by 2% lidocaine gel to reduce eye motion during the MRI examination. A receive-only surface coil (1.0-cm diameter) was centered on the left eye [18, 24]. 7T MRI data were acquired using several single spin-echo sequences (time to echo 13 ms, in-plane resolution $12 \times 12$ mm$^2$, matrix size $192 \times 192$, slice thickness 600 μm). Images were acquired at different TRs in the following order (number per time between repetitions in parentheses): repetition time (TR) 0.15 seconds (6), 3.50 seconds (1), 1.00 seconds (2), 1.90 seconds (1), 0.35 seconds (4), 2.70 seconds (1), 0.25 seconds (5), and 0.50 seconds (3). To compensate for reduced signal–noise ratios at shorter TRs, progressively more images were collected as the TR decreased. The present in-plane resolution in the central retina is sufficient for extracting meaningful layer-specific anatomical and functional data, as previously discussed [30].

## MRI data analysis

T1 data set of 23 images was first processed by registering (rigid body; STACKREG plugin, ImageJ, Rasband, W.S., ImageJ, U. S. National Institutes of Health, Bethesda, Maryland, USA, https://imagej.nih.gov/ij/, 1997–2016) and then averaging images with the same TRs in order to generate a stack of 8 images. These averaged images were then registered (rigid body) across TRs. R1 data were corrected for imperfect slice profile bias in the estimate of T1 as previously described (Chapter 18 in [31]). Briefly, normalizing to the shorter TR, some of the bias can be reduced giving a more precise estimate for T1. To achieve this normalization, we first apply a 3x3 Gaussian smoothing (performed three times) on only the TR 150 ms image to minimize noise and emphasize signal. The smoothed TR 150 ms image was then divided into the rest of the images in that T1 data set. Previously we found that this procedure minimized day-to-day variation in the R1 profile [14, 15]. R1 maps were calculated using the 7 normalized images via fitting to a three-parameter T1 equation ($y = a + b*(\exp(-c*TR))$, where a, b, and c are fitted parameters) on a pixel-by-pixel basis using R (v.2.9.0, R Development Core Team [2009]). R: A language and environment for statistical computing. R Foundation for Statistical Computing, Vienna, Austria. ISBN 3–900051–07–0) scripts developed in-house, and the minpack.lm package (v.1.1.1, Timur V. Elzhov and Katharine M. Mullen minpack.lm: R interface to the Levenberg-Marquardt nonlinear least-squares algorithm found in MINPACK. R package version 1.1–1). Our analysis separately compared superior and inferior values from ± 0.4 to 2 mm from the optic nerve head generated for each animal group.

Whole retinal thickness (μm) for each mouse was objectively determined from the MRI data using the "half-height method" wherein a border is determined via a computer algorithm based on the crossing point at the midpoint between the local minimum and maximum, as detailed elsewhere [32, 33]. The distance between two neighboring crossing-points thus represents an objectively defined retinal thickness. R1 profiles in each mouse were then normalized with 0% depth at the presumptive vitreoretinal border and 100% depth at the presumptive retina-choroid border. The present resolution is sufficient for extracting meaningful layer-specific anatomical and functional data, as previously discussed [11, 34].

## Statistical analysis

Data are presented as mean and 95% confidence intervals. A significance level of 0.05 was used for most tests, with interactions being tested using a significance level of 0.10 due to these tests

having less power. We used linear mixed models with the Kenward-Roger method for calculating degrees of freedom in SAS 9.4 (SAS software, Cary, NC, USA) to analyze both R1 and MRI thickness since both had repeated measures for each mouse. R1 is measured along retina depth, resulting in an MRI profile. We used restricted cubic splines to model and compare mouse-specific profiles between groups. The number of "windows" with a relationship between R1 and location (i.e., "knots") was initially evaluated separately for each group for any given analysis, and the Akaike and Schwarz Bayesian information criteria (AIC and BIC) were used to identify the model with the fewest knots needed to model all groups. Random coefficients for the intercept, side, the location-specific coefficients (cubic spline coefficients), and two-way interactions among these effects with mouse nested within strain and dose were also evaluated using AIC and BIC. Given the random spline coefficients included in all models, we used an unstructured covariance matrix for the random coefficients to account for associations in spline coefficients due to subject-specific profiles. The final model included the random coefficients for the intercept, side, depth, second, fourth and fifth knot, and knot*right coefficients. The model included the fixed effects of strain, DNP dose (0, 5, or 10 mg/kg), side of the ONL, location-specific values for the cubic splines, and four-way interactions among the main effects. The four-way interaction of strain, DNP dose, side, and depth was significant (p = 0.0678), indicating that MRI profiles depended on the specific combination of strain, dose and side. Mean profiles and comparisons of the mean profiles were generated using linear contrasts. Both linear and quadratic comparisons were evaluated among DNP doses since these two comparisons are independent and allow us to fully explain any differences among DNP doses.

For MRI thickness, the linear mixed model included the fixed effects of strain, DNP dose, side, and all interactions. The model also included a random intercept for mouse nested within strain and dose. No interactions were significant at the 0.10 level, leading to a final model that included only the main effects of strain, DNP dose, and side.

For OCT-based ELM-RPE thickness, we also used a linear mixed model that included the fixed effects of strain, DNP treatment, side, and all interactions among these fixed effects. We also included a random intercept for mouse nested within strain. Since DNP treatment was a within-subject effect for the OCT data, we also evaluated whether to include random coefficients for DNP and side using both AIC and BIC. Neither of these random effects were included since AIC/BIC was not reduced by adding these random coefficients to the model. The three-way interaction of strain, DNP treatment, and side and the two-way interaction of DNP treatment and side were not significant at the 0.10 level, leading to a final model that included all main effects and the two-way interactions of strain*DNP treat and strain*side. Since change in ELM-RPE thickness with DNP treatment did not differ between sides, we used the final model to generate estimates that averaged over side.

## Results

### Dark-adaptation + DNP studied by OCT

We first studied the impact of DNP using OCT studies in dark-adapted mice, when outer retina energy utilization and water removal capacity are normally greater than in the light [5, 28, 35, 36]. Fig 1 shows representative OCT images of the superior region of dark-adapted B6 and S6 mice given saline or DNP. ELM-RPE thinning is visibly evident in S6 mice but not in B6 mice. Quantitative summary of the OCT data did not differ by side and averages are presented in Fig 2. ELM-RPE thinning was statistically evident with DNP in S6 mice, but not in the B6 mice. This reduction in S6 mice was most apparent in Fig 2C which allows for an evaluation of whether the DNP-associated decrease is similar between the two strains. These results suggest

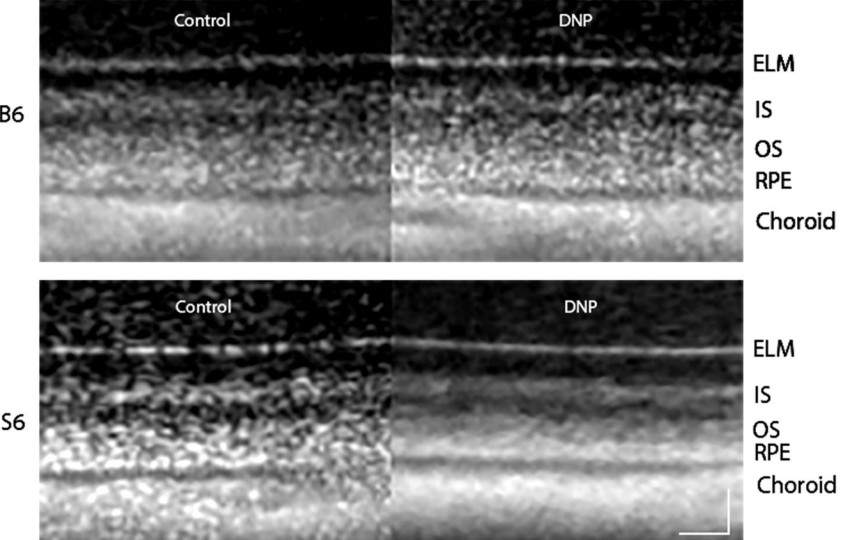

**Fig 1. Representative same-eye OCT images focused on the superior outer retina layers for dark-adapted B6 and S6 mice before (Control) and after DNP treatment.** Layer assignments are: ELM, external limiting membrane; IS, rod inner segment layer; OS, rod outer segment layer; RPE, retinal pigment epithelium, as reported earlier [37]. Scale bars, 20 μm.

that additional metabolic stimulation by systemic DNP, over and above that observed in the dark, produced a relatively more acidified subretinal space and greater water removal in S6, but not B6, mice consistent with the previously reported strain-dependent differential in mitochondrial reserve capacity [4].

## Dark-adaptation + DNP studied by MRI

Stimulating metabolism is expected to speed up the movement of electrons through the electron transport chain, allowing less time at each complex to produce paramagnetic free radicals.

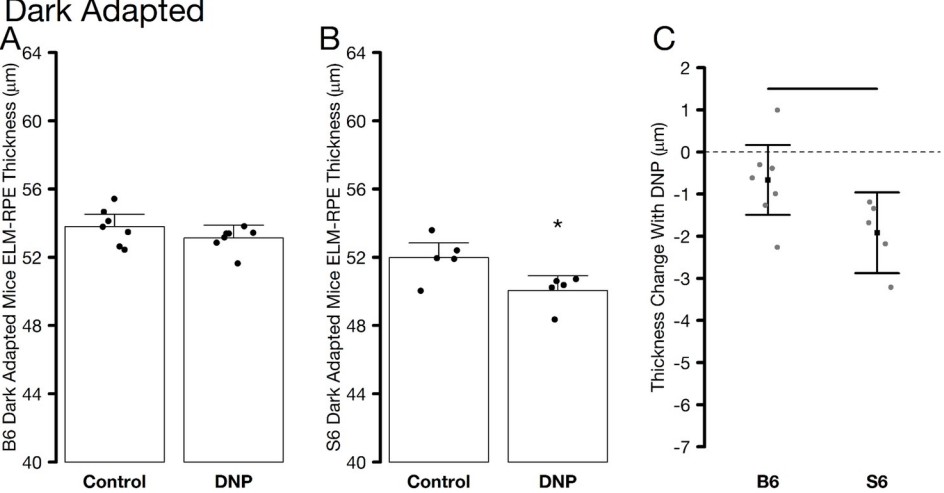

**Fig 2.** Summary of average ELM-RPE thickness measured from OCT images of dark-adapted control and DNP treated A) B6 and B) S6 mice. Individual data points (= number of eyes examined; one eye per mouse) represent the replicate average for each mouse to illustrate animal-to-animal variation. C) Paired differences to account for changes within mice; horizontal bar indicates significant differences between B6 and S6 groups. In all graphs, error bars represent 95% confidence intervals.

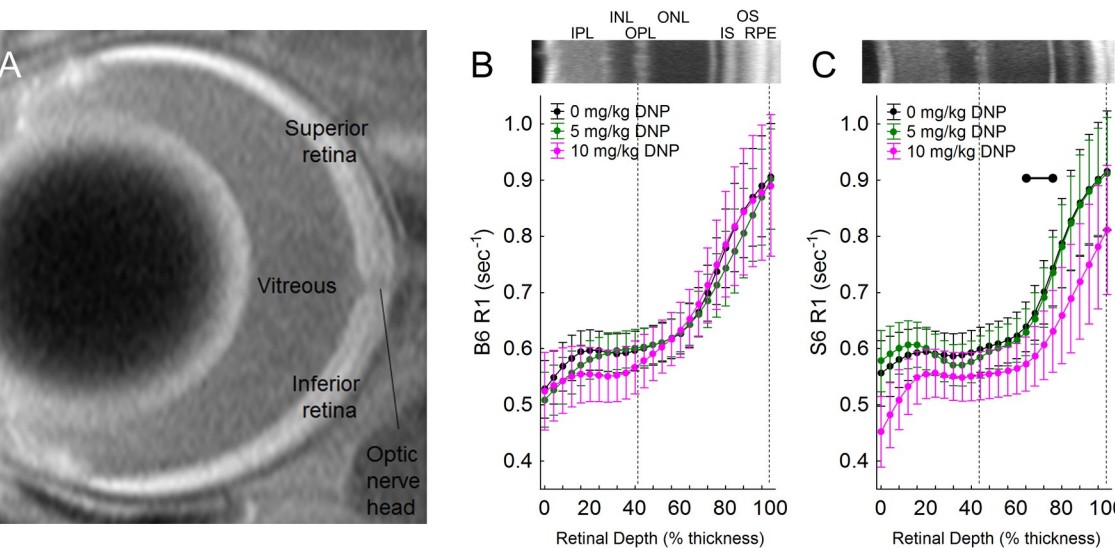

**Fig 3.** A) Representative high spatial resolution T1-weighted MRI. The retina shows a distinctive 4-layer pattern (bright, dark, bright, dark) on MRI, as previously reported; the anatomic identity of each dark / bright band has not been fully characterized [38]. The vitreous-retina and retina-choroid borders are visibly evident used as the only fiduciary alignment regions for comparing MRI and OCT data. MRI R1 profiles. Modeling results are shown for 1/T1 MRI profiles *in vivo* ~1 hr post DNP in *inferior retina* of dark-adapted. B) R1 vs depth (% of retinal thickness) for B6 mice after injection of either saline (0 mg/kg, black, n = 6), 5 mg/kg DNP (green, n = 10), or 10 mg/kg DNP (pink, n = 5), and C) R1 vs depth (% of retinal thickness) for S6 mice after injection of either saline (0 mg/kg, black, n = 7), 5 mg/kg DNP (green, n = 8), or 10 mg/kg DNP (pink, n = 10). Representative OCT images from inferior retina of B6 and S6 mice are presented above the profiles. Each 1/T1 data set was normalized to its TR 150 ms image ("Normalized 1/T1"), as detailed in the Methods section. OCT layer assignments (IPL, inner plexiform layer; INL, inner nuclear layer; OPL, outer plexiform layer; ONL, outer nuclear layer; IS, rod inner segment layer; OS, rod outer segment layer; RPE, retinal pigment epithelium) are as previously published [37]. Approximate location of two anatomic landmarks are indicated by dashed, vertical lines (i.e., anterior aspect of the OPL [left] and retina/choroid border [right]). Black range bar: Retinal depth range with significant difference (p < 0.05). Error bars represent 95% confidence intervals.

In turn, paramagnetic oxygen content is predicted to decrease. Thus, we next tested the prediction from the above OCT results that DNP will lower outer retina R1, an MRI parameter that is sensitive to levels of free radicals and oxygen, in dark-adapted S6 mice, but not B6 mice [6]. In dark-adapted S6 mice, a linear comparison of DNP was statistically significant in inferior retina (p = 0.001), indicating R1 decreased with DNP dose (Figs 3 and 4). This decrease was not observed in superior retina (Fig 5). Moreover, the DNP-evoked reduction in R1 was only significant in a limited region containing the mitochondrial-rich inner segment and posterior outer nuclear layers (64–84% depth from the vitreous-retina border) (Figs 3 and 4). On the other hand, B6 mice did not show a dose-dependent DNP effect in the inferior or superior retina at any location within the retina (Figs 3 and 5). The difference between the strains in the dose-dependent DNP effect was statistically significant at depths of 64–76% (a region just anterior to the inner segments, Fig 4) although a significant effect was noted in S6 mice in the inner segment region (Fig 4).

In S6 mice, a linear dose-dependent reduction in R1 was also found localized to the inferior retinal ganglion layer (0–4% depth) (not shown). DNP did not decrease R1 in the B6 inner, inferior retina (not shown). However, B6 mice did show a quadratic dose-dependent reduction in R1 in the non-photoreceptor region of the superior retina (0–20% depth), while S6 mice did not. The differences in these dose-dependent reductions in R1 were not statistically significant. Together, the DNP-assisted MRI R1 data provides independent *in vivo* support for interpretation of the above OCT data in terms of distinct mitochondrial energy ecosystems in S6 and B6 mice [4].

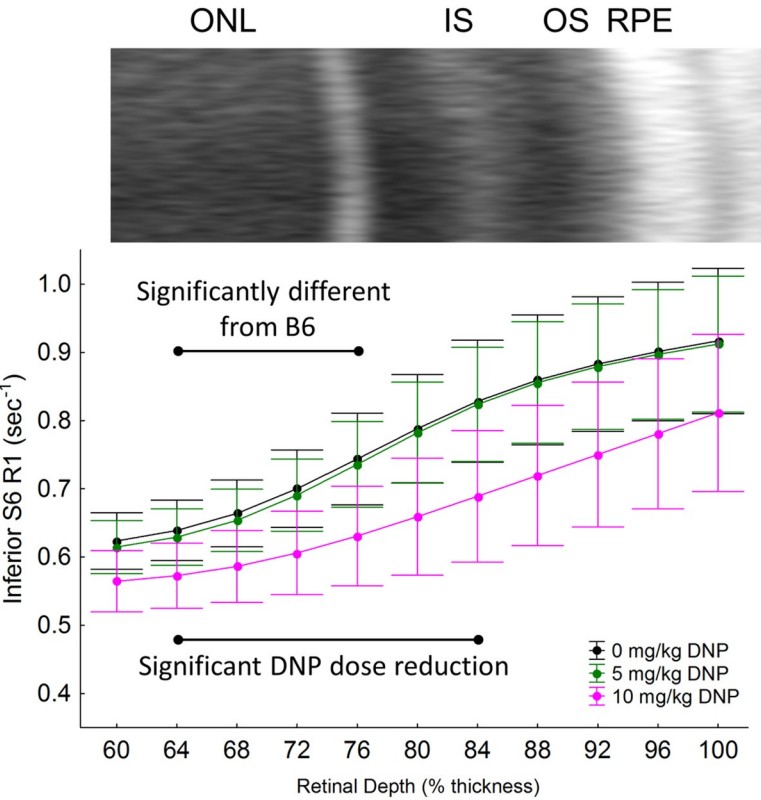

**Fig 4.** To highlight the localized differences in inferior retina of S6 mice, modeling results are replotted on an expanded scale for 1/T1 MRI profiles *in vivo* ~1 hr post DNP in *inferior retina* of dark-adapted mice after injection of either saline (0 mg/kg, black, n = 7), 5 mg/kg DNP (green, n = 8), or 10 mg/kg DNP (pink, n = 10). An expanded region of the representative OCT image from inferior retina of S6 mice is also shown above the profiles. Other conventions are as in Fig 4 and its legend. Black range bar: Retinal depth range with significant differences (p < 0.05) as indicated. Error bars represent 95% confidence intervals.

## Light-adaptation + DNP studied by OCT

To check if B6 mice are able to show morphological changes (i.e., water removal) in response to DNP, light-adapted mice were studied and ELM-RPE thickness measured. In light, rod energy production is relatively lower than that in the dark, and DNP metabolic stimulation would be expected in both mouse strains [28]. In the light, the OCT data again did not differ by inferior / superior side and were averaged. Light-adapted S6 and B6 mice both demonstrated ELM-RPE thinning with DNP (Fig 6). Thinning was greatest in the B6 mice (Fig 6C). These results support the notion that DNP can produce metabolic stimulation, with likely greater acidified subretinal space and associated increased water removal, in both S6 and B6 mice but in a light-dependent manner.

## Discussion

In this study, we demonstrate the utility of DNP-assisted imaging biomarkers for monitoring induced metabolic stimulation using doses reported to increase oxygen consumption in mammalian brain *in vivo* and in *ex vivo* studies [10, 23]. The short duration of DNP exposure herein (1 hr) likely precludes any substantial degradation of the uncoupled mitochondria by, for example, mitophagy [39]. Respiratory adaptation and culling of mitochondria generally takes chronic exposure to uncouplers [40]. The present OCT studies are sensitive to ELM-RPE

## Superior retina

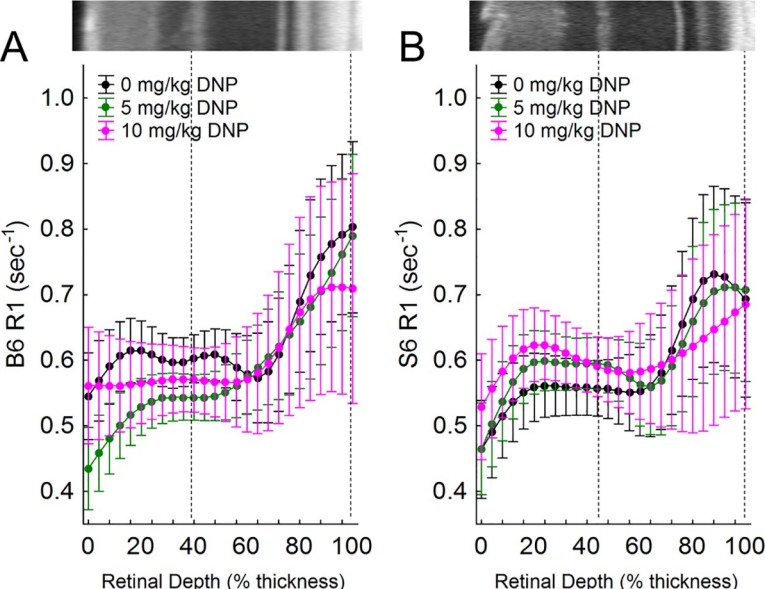

**Fig 5.** A) B6 after injection of either saline (0 mg/kg, black, n = 6), 5 mg/kg DNP (green, n = 10), or 10 mg/kg DNP (pink, n = 5), and C) S6 mice after injection of either saline (0 mg/kg, black, n = 7), 5 mg/kg DNP (green, n = 8), or 10 mg/kg DNP (pink, n = 10). Other conventions are as in Fig 4 and its legend.

hydration status which is modified by mitochondrial oxygen consumption [4]. Our working model is that when rods produce $CO_2$ and water, a pH-sensitive water removal co-transporter is upregulated on the RPE causing the ELM-RPE region to thin [4, 5, 11, 12, 41]. The strain-dependent response of ELM-RPE thinning patterns to DNP reported herein in dark, and supported by DNP-assisted R1 MRI measurements, appear consistent with our report of rod cells

## Light Adapted

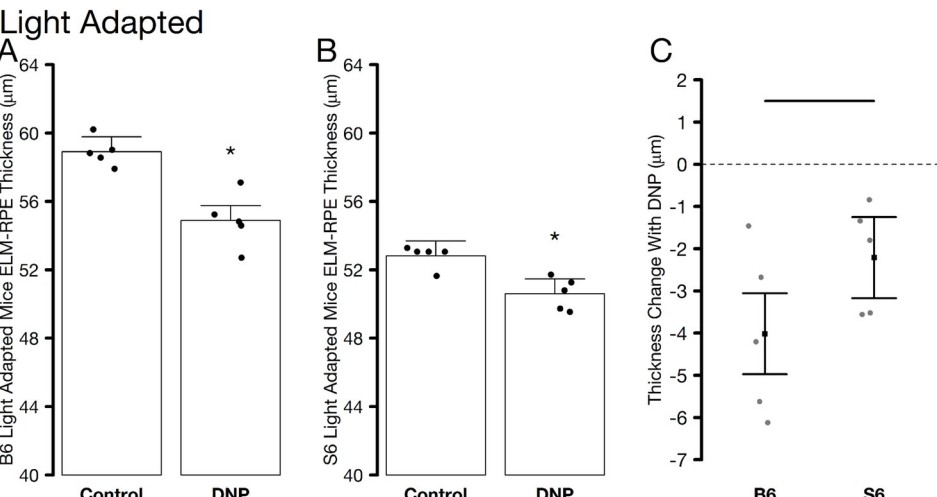

**Fig 6.** Summary of average ELM-RPE thickness measured from OCT images of light-adapted control and DNP treated A) B6 and B) S6 mice. C) Paired differences to account for changes within mice. All graphing conventions are as in Fig 2.

of S6 mice having a relatively greater mitochondrial reserve capacity than that of B6 mice based on *ex vivo* experiments [4, 42].

We considered the possibility that endogenous or high concentrations of uncouplers that reduce the protonmotive force might alter ATP synthase activity. However, as reviewed elsewhere, even relatively high, repeated doses of DNP do not alter ATP content or the ATP pool as measured *in vivo* using $^{31}$P magnetic resonance spectroscopy [43]. In the current study, we used relatively low doses, performed experiments in both dark-and light-adapted mice (which gave different results suggesting no evidence for DNP-induced damaging energetic limitation), and our measurements were done at acute time points. This latter point is germane since respiratory adaptation and culling of mitochondria is thought to take chronic exposure to uncouplers [40]. Nonetheless, future studies will assess this important question by direct measurement of ATP:ADP with and without DNP in the retina.

DNP might have potential off-target effects such as decreased cardiac output that could limit interpretation of actions selective to the retina. If systemic toxicity or off-target effects were a problem, one would expect to see global changes throughout the retina in our biomarkers in the two mouse strains. For example, if DNP stimulated a reduction in cardiac output it is anticipated that blood flow in the inner retina and choroid (the blood supply to the outer retina) would be lower which would affect inner retina thickness and R1; no such effect was seen in the S6 or B6 mice (Figs 1, 3 and 5). Instead, the effects of DNP were highly localized to the outer retina, were light-adaptation dependent, and were strain-specific consistent with an *ex vivo* report. An example of this is the localized impact of DNP to within the avascular outer retina with a dose-dependent R1 response noted in the inferior retina of dark-adapted S6 mice (Fig 4). In addition, a previous study examined similar doses of DNP as used herein and found increased brain oxygen consumption rates with stable vital signs in pigs [10]. Together, the above considerations argue against systemic or off-target contributions and support the notion that low doses of DNP can induce localized reductions in R1 in outer retina mitochondria, consistent with reduced free radicals and / or oxygen levels. However, The R1 of different ROS is not known at present and also we don't know the mixture of ROS that are impacting R1. Thus, it is unclear whether ROS or oxygen contribute more to the R1 signal.

A related potential drawback of DNP is that it has a limited safety index [9]. We note that the doses of 5–10 mg/kg DNP used herein are below the DNP LD50 of 35 to 72 mg/kg in mice [23, 44–46]. Systemic DNP given at a dose of 0.5 mg/kg or greater raised plasma alanine transaminase (ALT) and aspartate transaminase (AST) levels in rats suggesting the presence of liver damage [47]. Nonetheless, as was recently reviewed, low doses of DNP as used in this study did not appear to be toxic (see above) [9]. We note that the 10 mg/kg dose killed two out of the seven urethane-anesthetized B6 mice studied, none of the mice used for OCT (anesthetized with ketamine and xylazine), and the 20 mg/kg dose killed two urethane-anesthetized B6 mice in a preliminary experiment (data not shown). For this reason, doses higher than 10 mg/kg were not investigated in this study. It seems, based on the LD50 and other recently reviewed considerations that these fatalities do not reflect on DNP toxicity *per se* but rather a possible interaction between DNP and urethane [9]; the 20 mg/kg dose was not investigated in urethane-anesthetized S6 mice. Urethane is well studied and maintains systemic and neuronal physiology to levels found in conscious animals although its mode of action remains somewhat unclear [48–51]. Nonetheless, similar doses of DNP as used in this study were also used previously without systemic toxicity in ketamine-anesthetized pigs [10]. Future studies will investigate the usefulness of using other anesthetics that do not interact with DNP for the MRI experiments, as well as controlled-release oral formulations of DNP [52]. Alternatively, MP201 (a prodrug to DNP) also has a high safety index and no reported negative consequences in B6 mice at 80 mg/kg [52].

Physiologically, DNP is expected to stimulate an increase in oxygen consumption rate in mitochondria, particularly in the mitochondria-dense inner segment region but also in other regions such as the retinal pigment epithelium. The stimulation of respiration by DNP is also predicted to cause an increase in higher $CO_2$ and waste water production accompanied by increased removal of this acidified water in the ELM-RPE region. Water removal in dark-adapted mice is assisted by increased production of $CO_2$ and waste water leading to a more acidified environment that upregulates a co-transporter on the apical portion of the RPE, as demonstrated by comparing dark- and light-adapted retina and suggested previously by DNP-evoked reduction of edema in a nerve injury model [5, 8, 11–13]. Our observations that the ELM-RPE region gets smaller than normal with DNP in dark- and light-adapted B6 and S6 mice are in-line with a greater removal of acidified water linked with increased metabolism [4, 53]. In addition, an DNP-evoked increase in oxygen consumption will reduce levels of para-magnetic oxygen and free radical production that together provide a reasonable explanation for the observed reduction in R1 in S6 mice (Figs 1 and 2) [5, 8, 19, 21, 52, 54].

The present study also found a relatively greater ELM-RPE thinning in response to DNP in light-adapted B6 mice compared with S6 mice (Fig 6). These results support distinct strain-dependent mitochondrial energy ecosystems in the outer retina [4, 26, 42]. This might arise from, for example, having different set-points and / or regulation of the RPE water removal co-transporter in B6 than in S6 mice. More work is needed to unravel the different strain-dependent contributions from mitochondrial water production and RPE removal that under-lie this difference It is worth noting that S6 and B6 mice have a variety of genetic differences, including a RPE65 variant in S6 that increases rhodopsin regeneration rate compared to that in B6 mice [55], and a natural knock-out mutation in the nicotinamide (NAD) nucleotide transhydrogenase (Nnt) gene in B6 mice that can alter mitochondrial NAD(P) redox homeo-stasis [56]. Additional studies are needed to explore potential links these genetic mutations and the observed differences in OCT and MRI biomarkers [57, 58].

It is possible that DNP stimulation of metabolism will also generate heat that could decrease R1; the impact of heat on the ELM-RPE thickness is unclear [44]. However, body temperature raises with DNP are reported at doses greater than 25 mg/kg [47]. Also, a 5 mg/kg dose given daily does not cause weight loss in mice [10, 46]. We speculate that any production of heat by DNP does not exceed the heat-removal ability of the two circulatory beds of the retina [46, 59]. In addition, DNP-evoked increase in energy expenditure (i.e., heat generation) might be expected to increase retinal blood flow as has been observed in other organs, perhaps as a cool-ing mechanism [60]. However, an increase in blood flow would increase the inflow of fresh spins into the voxels and cause MRI R1 to increase in the vascular portions of the retina, and this was not seen in the inner retina of S6 and B6 mice (Figs 3 and 5); note that the photorecep-tor layer is avascular.

Intriguingly, OCT and MRI showed different spatial responses to DNP. For example, in S6 mice, DNP affected the ELM-RPE region in both inferior and superior retina, whereas only R1 in the inferior retina was affected. We speculate that this difference reflects a relatively lower statistical power in the MRI data due to a combination of, for example, the cross-sectional experimental design used instead of the paired design in the OCT study, and the lower spatial resolution of MRI (21 μm) compared to that of OCT (1.6 μm). Unfortunately, a paired design is not feasible for the MRI experiment because of the use of the terminal anesthetic urethane. Alternative anesthetics, such as isoflurane or ketamine/xylazine, have limited use for the retinal MRI studies. The MRI eye coil occupies the same physical space as the isoflurane nose cone so both cannot be used together. Ketamine / xyalzine provides anesthesia for only ~30 min which is too short for a typical MRI examination which lasts ~ 1 hr. Presenting the retina with a higher concentration of DNP might improve the detection sensitivity of the MRI experiment.

More work is needed to determine if, for example, direct intraocular injection of DNP, or alternative formulations of DNP (see below) would be useful in this regard [47, 61]. Nonetheless, our primary observation that DNP elicits changes in both MRI and OCT biomarkers in the outer retina in S6 mice relative to B6 mice supports our working hypothesis [4].

In summary, the present results support continued investigation of ELM-RPE OCT thickness and R1 MRI measurements as useful approaches for monitoring the impact of mitochondrial protonophores for evaluating *in vivo* the rod photoreceptor mitochondrial energy ecosystem [9, 46, 47, 52, 61]. The development of new approaches for *in vivo* imaging of mitochondrial uncoupling response maps raises the possibility for promising future applications for evaluating mitochondrial protonophore treatment in neurodegenerative disease [46, 47, 52, 61].

## Supporting information

**S1 Fig. Data for Fig 2.**
(ZIP)

**S2 Fig. Data for Fig 3.**
(ZIP)

**S1 Data and Code. Entire data sets for both MRI and OCT measurements, along with SAS code for the final model.**
(ZIP)

## Acknowledgments

This research was gratefully supported by the National Institutes of Health [RO1 EY026584 (BAB) and R01 AG058171 (BAB)], Kentucky Spinal Cord and Head Injury Research Trust (KSCHIRT) Grant 14-13A and VA Merit Award 1I01BX003405 (PGS), NIH intramural Research Programs EY000503 and EY000530 to HQ, NEI Core Grant P30 EY04068, and an unrestricted grant from Research to Prevent Blindness (Kresge Eye Institute, BAB), a Fight for Sight Summer Student Fellowship (CR), and Wayne State University School of Medicine Medical Student Summer Research Fellowships (HKO and JJ).

## Author Contributions

**Conceptualization:** Bruce A. Berkowitz, W. Brad Hubbard, Patrick G. Sullivan, Haohua Qian.

**Formal analysis:** Bruce A. Berkowitz, Robert H. Podolsky, Karen Lins Childers, Yichao Li, Haohua Qian.

**Funding acquisition:** Bruce A. Berkowitz, Haohua Qian.

**Investigation:** Bruce A. Berkowitz, Hailey K. Olds, Collin Richards, Joydip Joy, Shasha Gao, Robin Roberts.

**Methodology:** Patrick G. Sullivan.

**Project administration:** Bruce A. Berkowitz.

**Resources:** W. Brad Hubbard, Patrick G. Sullivan.

**Software:** Tilman Rosales.

**Supervision:** Bruce A. Berkowitz, Haohua Qian, Robin Roberts.

**Writing – original draft:** Bruce A. Berkowitz, Robert H. Podolsky, W. Brad Hubbard, Patrick G. Sullivan, Haohua Qian.

**Writing – review & editing:** Bruce A. Berkowitz, Hailey K. Olds, Collin Richards, Joydip Joy, Tilman Rosales, Robert H. Podolsky, Karen Lins Childers, W. Brad Hubbard, Patrick G. Sullivan, Haohua Qian, Robin Roberts.

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
