## [Decision Letter · Decision Letter 0]

1 Oct 2019

PONE-D-19-24762

Novel Imaging Biomarkers for Mapping the Impact of Mild Mitochondrial Uncoupling in the Outer Retina In Vivo

PLOS ONE

Dear Dr. Berkowitz (Bruce),

Thank you for submitting your manuscript to PLOS ONE. After careful consideration, we feel that it has merit but does not fully meet PLOS ONE’s publication criteria as it currently stands. Therefore, we invite you to submit a revised version of the manuscript that addresses the points raised during the review process.

Please include  light adapted retinas in your study, and consider the impact of ATP depletion as a consequence of dinitrophenol treatment.

We would appreciate receiving your revised manuscript by Nov 15 2019 11:59PM. To enhance the reproducibility of your results, we recommend that if applicable you deposit your laboratory protocols in protocols.io, where a protocol can be assigned its own identifier (DOI) such that it can be cited independently in the future. For instructions see: http://journals.plos.org/plosone/s/submission-guidelines#loc-laboratory-protocols

We look forward to receiving your revised manuscript.

Best wishes for 5780,

Al Lewin

Academic Editor

PLOS ONE

**Journal Requirements:**

2. Thank you for including your ethics statement: "All animals were treated in accordance with the National Institutes of Health Guide for the Care and Use of Laboratory Animals, the Association for Research in Vision and Ophthalmology Statement for the Use of Animals in Ophthalmic and Vision Research, and Institutional Animal and Care Use Committee authorizations at WSU.

a. Please amend your current ethics statement to include the full name of the ethics committee that approved your specific study and confirm that your named ethics committee specifically approved this study.

For additional information about PLOS ONE submissions requirements for animal ethics, please refer to http://journals.plos.org/plosone/s/submission-guidelines#loc-animal-research

**Comments to the Author**

1. Is the manuscript technically sound, and do the data support the conclusions?

Reviewer #1: Partly

Reviewer #2: Yes

2. Has the statistical analysis been performed appropriately and rigorously? 

Reviewer #1: Yes

Reviewer #2: Yes

3. Have the authors made all data underlying the findings in their manuscript fully available?

Reviewer #1: Yes

Reviewer #2: Yes

4. Is the manuscript presented in an intelligible fashion and written in standard English?

Reviewer #1: Yes

Reviewer #2: Yes

5. Review Comments to the Author

Reviewer #1: The manuscript under review by Bruce Berkowitz and collaborators describes potentially useful MRI and OCT determinations to evaluate the effects of systemic administration of an oxidative phosphorylation uncoupler on the retina. The article is clearly written and the data is well presented and carefully analyzed. However, the experimental design and interpretation of the data need to be reassessed in view of the following considerations:

1) Only dark adapted retinas were analyzed. The justification for this is based on the accepted view (refs. 23-25) that oxidative phosphorylation in the retina is more active in the dark than under light because of a faster ATP turnover. However, this implies that mitochondrial reserve capacity is reduced in the dark, that is, basal respiration will be closer to its maximal possible rate in the dark than under light adapted conditions. This limits the effect of uncoupling by administration of agents such as DNP, given that respiration rate is almost completely maxed out even in the absence of endogenous uncoupler. Inclusion in the study of light adapted retinas should render the effects of uncoupling much more evident, while still showing strain-specific quantitative differences.

2) Ex-vivo basal and maximal respiration (used to calculate mitochondrial reserve capacity) comparing S6 and B6 retinas has only been assessed under light adapted conditions (ref. 4), but not after dark adaptation. Water content and retinal thickness was found to be different between the two strains only under light (but not dark) adaptation (ref. 4). This also highlights the need to perform the reported experiments under light adapted conditions.

3) Mild uncoupling should not be considered to simply decrease oxygen concentration and acidity by virtue of an increased mitochondrial oxygen consumption and CO2 generation; its more important (and undesirable) effect is likely the decrease in the rate of ATP synthesis by dissipating the protonmotive force across the mitochondrial membrane, especially in photoreceptors (and more critically in the dark because of a higher ATP demand), where little mitochondrial reserve capacity exists. Is the content of high energy phosphates (i.e. creatine phosphate and ATP) changing upon addition of DNP? These have been monitored in other tissues by NMR (see Balaban, RS et al. (1986) Science 232:1121-3), or could also be quantified by mass spectrometry (metabolomics approach) or even by biochemical methods (although less reliably). If ATP/creatine phosphate content does change, is the energy compromise the real cause of the change in water mobility? Uncoupled mitochondria are usually targeted for degradation by quality control mechanisms such as mitophagy (see Georgakopoulos ND et al. (2017) Nat Chem Biol 13:136-146, for example). Is this happening in dark adapted photoreceptors at the DNP concentrations used? What would be the consequences of such changes on the imaging biomarkers presented in this paper? Even if the DNP concentrations used in this paper have been validated as non-toxic for the brain (ref. 10), it could well be that cells with much lower reserve capacity than brain neurons, such as photoreceptors, are being subjected to a damaging energetic limitation. Therefore, experiments showing that the so-called "mild" uncoupling employed in the present study is not compromising the energetic status of the retina should be included.

4) As stated by the authors, the MRI R1 is sensitive to both oxygen and ROS concentrations. These do not necessarily go in the same direction (see for instance Quarrie R et al. (2014) Am J Physiol Heart Circ Physiol. 307:H996-H1004). Which molecules contribute more to the R1 signal? Can they be distinguished by other methods? Related to this, NADPH is important for detoxifying ROS in mitochondria (see Francisco A et al. (2018) J Neurochem. 2018 147:663-677). Is the B6 strain used in the present study the same as the C57BL/6J strain that has a truncated version of the Nnt gene that codes for mitochondrial nicotinamide nucleotide transhydrogenase? If so, could this be the reason in the different patterns reported between the S6 and B6 strain?

Reviewer #2: This is an important study from a group that has developed unique MRI-based methods and used them together with OCT to quantify and characterize distributions of O2, free radicals and water in mouse eyes. This study focuses on the effects of dinitrophenol on these distributions. Previous studies from this lab showed metabolic differences between two mouse strains so this report also includes a comparison of the effects of DNP on retinas from those two mouse strains. The study showed that DNP causes thinning and lowering of O2 levels in specific parts of the retinas and that this effect occurs to a greater extent in the S6 strain than in the B6 strain. The molecular interpretation of the findings is not yet definitive, but the authors do show how the methodology is fairly robust and that it can be used to report effects of metabolic perturbations.

Specific comments/suggestions:

1. line 84 - "...oxygen consumption rates are greater in S6 mice than in B6 mice..." Is this correct? - my reading of Fig. 1 in ref 4 was that O2 consumption is faster in B6 than in S6. Please correct or clarify.

2. This study reports effects only on dark-adapted animals whereas other reports from this group also included effects of light adaptation. Mitochondria from dark-adapted animals may be more active because of the increased demand for ATP synthesis in darkness than in light. Based on that, I would predict that effects of DNP might be more pronounced in light than they would be in darkness. Based on previous reports there could a greater fold increase in O2 consumption induced by DNP in light compared to darkness, there also could be a greater fold increase in water production induced by DNP in light compared to dark. Also, the authors reported previously that there is a hyporeflective band in OCT in light that is not present in darkness, so it I think it would be very relevant in this report to determine if there are effects of DNP on that feature. Such an experiment may help to define whether the hyporeflective band arises from a functional effect such as phototransduction of if instead it is caused by a metabolic effect (mitochondria may become more active with DNP, which would simulate their increased activity in darkness.

6. PLOS authors have the option to publish the peer review history of their article (what does this mean?). If published, this will include your full peer review and any attached files.

Reviewer #1: Yes: Raul Covian

Reviewer #2: Yes: James Bryant Hurley

---

## [Author Response · Author response to Decision Letter 0]

22 Nov 2019

Reviewer #1: The manuscript under review by Bruce Berkowitz and collaborators describes potentially useful MRI and OCT determinations to evaluate the effects of systemic administration of an oxidative phosphorylation uncoupler on the retina. The article is clearly written and the data is well presented and carefully analyzed. However, the experimental design and interpretation of the data need to be reassessed in view of the following considerations:

1) Only dark adapted retinas were analyzed. The justification for this is based on the accepted view (refs. 23-25) that oxidative phosphorylation in the retina is more active in the dark than under light because of a faster ATP turnover. However, this implies that mitochondrial reserve capacity is reduced in the dark, that is, basal respiration will be closer to its maximal possible rate in the dark than under light adapted conditions. This limits the effect of uncoupling by administration of agents such as DNP, given that respiration rate is almost completely maxed out even in the absence of endogenous uncoupler. Inclusion in the study of light adapted retinas should render the effects of uncoupling much more evident, while still showing strain-specific quantitative differences.

Response: We tested the reviewer’s hypothesis, suggested by results in the literature collected on post-mortem retina, that light-adapted retina should render the effects of uncoupling much more evident while still showing strain-specific differences. The new ELM-RPE thickness data in the revision in light-adapted mice supported this hypothesis and the Results section, and Discussion paragraphs 1 and 6, are modified accordingly. 

2) Ex-vivo basal and maximal respiration (used to calculate mitochondrial reserve capacity) comparing S6 and B6 retinas has only been assessed under light adapted conditions (ref. 4), but not after dark adaptation. Water content and retinal thickness was found to be different between the two strains only under light (but not dark) adaptation (ref. 4) also highlights the need to perform the reported experiments under light adapted conditions.

Response: Please see response to 1) above. 

3) Mild uncoupling should not be considered to simply decrease oxygen concentration and acidity by virtue of an increased mitochondrial oxygen consumption and CO2 generation; its more important (and undesirable) effect is likely the decrease in the rate of ATP synthesis by dissipating the protonmotive force across the mitochondrial membrane, especially in photoreceptors (and more critically in the dark because of a higher ATP demand), where little mitochondrial reserve capacity exists. Is the content of high energy phosphates (i.e. creatine phosphate and ATP) changing upon addition of DNP? These have been monitored in other tissues by NMR (see Balaban, RS et al. (1986) Science 232:1121-3), or could also be quantified by mass spectrometry (metabolomics approach) or even by biochemical methods (although less reliably). If ATP/creatine phosphate content does change, is the energy compromise the real cause of the change in water mobility? 

Response: This comment is interesting and may have implications in states of endogenous or high concentrations of uncouplers that reduce the protonmotive force enough to alter ATP synthase activity. However, as reviewed and summarized by Geisler (PMID: 30909602), studies assessing this question have demonstrated that even relatively high, repeated doses of DNP do not alter ATP content or the ATP pool as measured using 31P magnetic resonance spectroscopy. In the current study we used relatively low doses and our measurements were done in acute time points. Regardless, future studies will be able to assess this important question by direct measurement of ATP:ADP with and without DNP in the retina. A new second paragraph in the Discussion was added.

4) Uncoupled mitochondria are usually targeted for degradation by quality control mechanisms such as mitophagy (see Georgakopoulos ND et al. (2017) Nat Chem Biol 13:136-146, for example). Is this happening in dark adapted photoreceptors at the DNP concentrations used? What would be the consequences of such changes on the imaging biomarkers presented in this paper?

Response: This topic is very important but is not germane to the current study given the short duration of the experiments. Respiratory adaptation and culling of mitochondria generally takes chronic exposure to uncouplers (PMID: 21187326). The first paragraph of the Discussion has been modified accordingly.

5) Even if the DNP concentrations used in this paper have been validated as non-toxic for the brain (ref. 10), it could well be that cells with much lower reserve capacity than brain neurons, such as photoreceptors, are being subjected to a damaging energetic limitation. Therefore, experiments showing that the so-called "mild" uncoupling employed in the present study is not compromising the energetic status of the retina should be included.

Response: We have modified the 1st, 2nd and 3rd paragraphs of the discussion to indicate that the results in the light demonstrate that B6 mice are able to react to DNP (i.e., no evidence for damaging energetic limitations), and thus further support the above interpretation that DNP given to dark-adapted mice reflects changes to oxygen consumption that can be attributed to mitochondrial reserve capacity in vivo, not due to energetic limitations.

6) As stated by the authors, the MRI R1 is sensitive to both oxygen and ROS concentrations. These do not necessarily go in the same direction (see for instance Quarrie R et al. (2014) Am J Physiol Heart Circ Physiol. 307:H996-H1004). Which molecules contribute more to the R1 signal? Can they be distinguished by other methods? Related to this, NADPH is important for detoxifying ROS in mitochondria (see Francisco A et al. (2018) J Neurochem. 2018 147:663-677). Is the B6 strain used in the present study the same as the C57BL/6J strain that has a truncated version of the Nnt gene that codes for mitochondrial nicotinamide nucleotide transhydrogenase? If so, could this be the reason in the different patterns reported between the S6 and B6 strain?

Response: Our results from retina in vivo are consistent with the results from the Quarrie reference in cardiac tissue ex vivo which showed that pharmacologically increasing H+ leak decreases mitochondrial ROS production if measured before an episode of ischemia-reperfusion (citation added to the introduction). We also note that both oxygen and free radicals are paramagnetic and thus at their low in vivo concentrations will alter R1 in the same direction (but to different extents). The R1 of different ROS is not known at present and also we don’t know the mixture of ROS that are impacting R1. Thus, it is unclear whether ROS or oxygen contribute more to the R1 signal. The third paragraph has been modified accordingly. We had somehow missed the important information about B6 mice lacking the Nnt gene (thank you!) and have added information to the 6th paragraph of the discussion.

Reviewer #2: This is an important study from a group that has developed unique MRI-based methods and used them together with OCT to quantify and characterize distributions of O2, free radicals and water in mouse eyes. This study focuses on the effects of dinitrophenol on these distributions. Previous studies from this lab showed metabolic differences between two mouse strains so this report also includes a comparison of the effects of DNP on retinas from those two mouse strains. The study showed that DNP causes thinning and lowering of O2 levels in specific parts of the retinas and that this effect occurs to a greater extent in the S6 strain than in the B6 strain. The molecular interpretation of the findings is not yet definitive, but the authors do show how the methodology is fairly robust and that it can be used to report effects of metabolic perturbations.

Specific comments/suggestions:

1. line 84 - "...oxygen consumption rates are greater in S6 mice than in B6 mice..." Is this correct? - my reading of Fig. 1 in ref 4 was that O2 consumption is faster in B6 than in S6. Please correct or clarify. 

Response: We apologize for the oversight, it is now fixed.

2. This study reports effects only on dark-adapted animals whereas other reports from this group also included effects of light adaptation. Mitochondria from dark-adapted animals may be more active because of the increased demand for ATP synthesis in darkness than in light. Based on that, I would predict that effects of DNP might be more pronounced in light than they would be in darkness. Based on previous reports there could a greater fold increase in O2 consumption induced by DNP in light compared to darkness, there also could be a greater fold increase in water production induced by DNP in light compared to dark. 

Response: We tested the reviewer’s hypothesis, suggested by results in the literature collected on post-mortem retina, that light-adapted retina should render the effects of uncoupling much more evident. The new ELM-RPE thickness data in the revision in light-adapted mice supported this hypothesis and the Results section, and Discussion paragraphs 1 and 6, are modified accordingly. 

3. Also, the authors reported previously that there is a hyporeflective band in OCT in light that is not present in darkness, so it I think it would be very relevant in this report to determine if there are effects of DNP on that feature. Such an experiment may help to define whether the hyporeflective band arises from a functional effect such as phototransduction of if instead it is caused by a metabolic effect (mitochondria may become more active with DNP, which would simulate their increased activity in darkness.

Response: We agree that the hyporeflective band that appears under light-adapted conditions is important, but it is a phenomena that is somewhat less understood than the regulation of the ELM-RPE thickness. For this reason, our analysis of the hyporeflective band is a focus in another manuscript in preparation.

---

## [Decision Letter · Decision Letter 1]

9 Dec 2019

Novel Imaging Biomarkers for Mapping the Impact of Mild Mitochondrial Uncoupling in the Outer Retina In Vivo

PONE-D-19-24762R1

Dear Dr. Berkowitz (Bruce),

We are pleased to inform you that your manuscript has been judged scientifically suitable for publication and will be formally accepted for publication once it complies with all outstanding technical requirements.

With kind regards,

Al Lewin, Ph.D.

Section Editor

PLOS ONE

Additional Editor Comments (optional):

Reviewers' comments:

Reviewer's Responses to Questions

**Comments to the Author**

1. If the authors have adequately addressed your comments raised in a previous round of review and you feel that this manuscript is now acceptable for publication, you may indicate that here to bypass the “Comments to the Author” section, enter your conflict of interest statement in the “Confidential to Editor” section, and submit your "Accept" recommendation.

Reviewer #1: All comments have been addressed

Reviewer #2: All comments have been addressed

2. Is the manuscript technically sound, and do the data support the conclusions?

Reviewer #1: Yes

Reviewer #2: Yes

3. Has the statistical analysis been performed appropriately and rigorously? 

Reviewer #1: Yes

Reviewer #2: Yes

4. Have the authors made all data underlying the findings in their manuscript fully available?

Reviewer #1: Yes

Reviewer #2: Yes

5. Is the manuscript presented in an intelligible fashion and written in standard English?

Reviewer #1: Yes

Reviewer #2: Yes

6. Review Comments to the Author

Reviewer #1: (No Response)

Reviewer #2: (No Response)

7. PLOS authors have the option to publish the peer review history of their article (what does this mean?). If published, this will include your full peer review and any attached files.

Reviewer #1: No

Reviewer #2: Yes: James B Hurley

---

## [Editor Report · Acceptance letter]

23 Dec 2019

PONE-D-19-24762R1 

Novel Imaging Biomarkers for Mapping the Impact of Mild Mitochondrial Uncoupling in the Outer Retina In Vivo 

Dear Dr. Berkowitz:

I am pleased to inform you that your manuscript has been deemed suitable for publication in PLOS ONE. Congratulations! Your manuscript is now with our production department. 

With kind regards,

on behalf of

Dr. Alfred S Lewin 

Section Editor

PLOS ONE